# Galectins in Cancer and the Microenvironment: Functional Roles, Therapeutic Developments, and Perspectives

**DOI:** 10.3390/biomedicines9091159

**Published:** 2021-09-04

**Authors:** Chien-Hsiu Li, Yu-Chan Chang, Ming-Hsien Chan, Yi-Fang Yang, Shu-Mei Liang, Michael Hsiao

**Affiliations:** 1Genomics Research Center, Academia Sinica, Taipei 115, Taiwan; dicknivek@icloud.com (C.-H.L.); ahsien0718@gmail.com (M.-H.C.); 2Department of Biomedical Imaging and Radiological Sciences, National Yang Ming Chiao Tung University, Taipei 112, Taiwan; jameskobe0@gmail.com; 3Department of Medical Education and Research, Kaohsiung Veterans General Hospital, Kaohsiung 81362, Taiwan; yvonne845040@gmail.com; 4Agricultural Biotechnology Research Center, Academia Sinica, Taipei 115, Taiwan; 5Department of Biochemistry, College of Medicine, Kaohsiung Medical University, Kaohsiung 80708, Taiwan

**Keywords:** galectin, metabolism, tumor microenvironment, cancer

## Abstract

Changes in cell growth and metabolism are affected by the surrounding environmental factors to adapt to the cell’s most appropriate growth model. However, abnormal cell metabolism is correlated with the occurrence of many diseases and is accompanied by changes in galectin (Gal) performance. Gals were found to be some of the master regulators of cell–cell interactions that reconstruct the microenvironment, and disordered expression of Gals is associated with multiple human metabolic-related diseases including cancer development. Cancer cells can interact with surrounding cells through Gals to create more suitable conditions that promote cancer cell aggressiveness. In this review, we organize the current understanding of Gals in a systematic way to dissect Gals’ effect on human disease, including how Gals’ dysregulated expression affects the tumor microenvironment’s metabolism and elucidating the mechanisms involved in Gal-mediated diseases. This information may shed light on a more precise understanding of how Gals regulate cell biology and facilitate the development of more effective therapeutic strategies for cancer treatment by targeting the Gal family.

## 1. Introduction

In 1977, Halina Den and coworkers performed β-D-galectinoside-specific lectin isolation from chickens [1], introducing the roles of Gal in cell biology. Until now, the versatility and controversial roles of galectins have remained diverse according to an increasing number of studies [2]. How specific galectins connect and remodel their ordinary function to disease development remains an elusive question. In this review, we organize the current understanding of galectins, from their fundamental roles to disease development, and their druggable potential for readers to appreciate their importance.

## 2. Classification and Distribution of Galectins 

Gal is one of the carbohydrate-binding proteins that belongs to the animal lectin family. Gals are ubiquitous in the cytosol, nucleus, plasma membrane, and extracellular regions of cells through binding to glycans, which contain lactose or N-acetyllactosamine (LacNAc; Galβ1-4GlcNAc), via van der Waals interactions [3]. Gals are transcribed and translated into proteins by *LGALS* genes. Their main structure is divided into the N-terminal domain (NTD), which is composed of 12 amino acids containing serine phosphorylation [4]. The middle comprises a proline–glycine-rich domain, and the β-sheet consists of approximately 130 amino acids that form a highly conserved, small, and soluble structure composed of the functional carbohydrate recognition domain (CRD), which has an affinity for binding to the β-galactoside and carbohydrates that further classify Gal family proteins through CRDs [4,5]. Currently, Gals are divided into three groups according to their different structures. (1) Dimeric Gals are composed of the same subunit as the single CRD component, and the prototype includes Gal-1, Gal-2, Gal-5, Gal-7, Gal-10, Gal-11, Gal-13, Gal-14, and Gal-15. (2) Tandem-repeat Gals include Gal-4, Gal-6, Gal-8, Gal-9, and Gal-12 and are composed of at least two carbohydrate CRDs that covalently interact. (3) Finally, Gals can present as monomers or multivalent chimera types based on NTD self-oligomerization (Gal-3) (Figure 1A). In addition, Gal can undergo polymerization through non-covalent bonds [6], which results in different carbohydrate associations and allows presentation in different subcellular components and tissues [7]. The conformational change in the subunits may affect Gal’s interactions with different partners through either carbohydrate-dependent binding or carbohydrate-independent binding to perform diverse functions [8,9]. Furthermore, posttranscriptional and posttranslational modifications affect the multiple isoforms of Gal that may decide subcellular distribution and protein stability. For example, Haddad et al. identified that *LGALS3* has an alternative reading frame called the *GALIG* gene, which can translate into mitogaligin and cytogaligin [10]. Apart from *LGALS5*, *LGALS6*, *LGALS11*, *LGALS14*, and *LGALS15* are expressed in other species, such as sheep and goats. There are currently approximately eleven *LGALS* whose subcellular location and expression in the human body have been characterized, as depicted in Figure 1B (Table 1).

## 3. Gal Functions in Cell Biology

To date, these Gals have been found to regulate different cellular functions, including the interaction of galactoside ligands with different proteins in embryonic development, inflammation, immune response, metabolic disease, premRNA splicing, cell cycle, motility, survival, organ fibrosis, and cancer development (Figure 2) [9,10,11,12,13,14,15].

### 3.1. Embryonic Development

Gal families are involved in chicken lens development during embryogenesis, i.e., Gal-1, Gal-3, and Gal-8 [16]. Increased expression of *LGALS1* may regulate the differentiation of human embryonic stem cells into pancreatic β cells [17]. Motohashi et al. reported that Gal-1 was highly expressed and enhanced neural crest generation in mouse embryonic stem cells [18]. Tang and others have shown that Gal-1 regulates trophoblast stem cell differentiation [11,19]. Gal-3 has been observed in myeloid cells, which can differentiate into macrophages in the developing lung and kidney [20,21].

### 3.2. Immunol Responses

During inflammation, Gal-1 induces neutrophil and T cell apoptosis [22,23]. Inflammatory bowel disease patients produce high serum levels of Gal-1 and Gal-3 [24]. Moreover, recombinant Gal-1 attenuates anti-ovalbumin glucose immunoglobulin E and interleukin to alleviate allergic airway inflammation [12]. Recombinant Gal-9 reverses lipopolysaccharide (LPS)-induced preeclampsia by promoting M2 macrophage polarization [25]. Interestingly, apart from Gal-1, Gal-3, a promoter during autoimmune cholangitis, is induced by activation of IL-1*β* and the NLRP3 inflammasome [26]. However, Gal-13 promotes neutrophil function by inducing reactive oxygen species (ROS), hepatocyte growth factor (HGF), matrix metalloproteinase 9 (MMP9), and programmed death-ligand 1 (PD-L1) during pregnancy [27].

### 3.3. Metabolic Processes 

In addition, Gal expression is related to metabolic disorders. Glucocorticoid treatment can reverse diabetic retinopathy-induced Gal-1 expression in hypoxia [13]. During pregnancy, Gal-2 was observed in a metabolic disorder that caused gestational diabetes mellitus [28]. Increased serum Gal-3 and Gal-4 expression was observed in diabetes patients [29,30], and levels of Gal-3 were correlated with cardiovascular events in type 2 diabetes mellitus patients [31,32,33]. In a type 1 diabetes model, targeting Gal-3 decreased pro-inflammatory cytokine production [34]. Damage to pancreatic β cells induced by Gal-3 was present in both type 1 and 2 diabetes [34,35]. Sun et al. found that Gal-3 mediates high glucose-induced cardiomyocyte injury by regulating NADPH oxidase [36]. Moreover, diabetes mellitus is accompanied by organ fibrosis, such as in cardiac and lung tissue [15,37]. Hernández-Romero et al. observed that Gal-3 is involved in diabetes-induced atrial fibrillation [38]. In agreement with other reports, Wu et al. showed that diabetes mellitus-induced atrial fibrillation was accompanied by increased Gal-3 expression [39]. Similar observations were described by Al-Obaidi et al., who reported increased Gal-1 expression in both type 1 and 2 diabetes that regulates hyperglycemia-induced renal fibrosis [40]. Altogether, current observations suggest that Gal participates in diverse cellular pathways; therefore, it is crucial to understand and elucidate Gal diversity.

## 4. Abnormal Regulation of Galectin in Cancer Progression

Based on the expression of Gals and their different functions in cells as described above, it is predictable that Gal is involved in tumor progression. The correlation between Gal and survival outcomes from The Cancer Genome Atlas (TCGA) is shown in Table 2. In pan-cancer, we found that different Gals will have coordinated and redundant effects. Studies have also shown that Gal has a multifunctional effect in certain cancers (Table 3). For example, Califice et al. demonstrated that cytoplasmic Gal-3 promotes prostate cancer motility, proliferation, and angiogenesis. However, it inhibits prostate cancer in the nucleus [14]. Similarly, distinct roles were observed in Gal-9 isoforms, including the C-terminus and N-terminus of the Gal-9 CRD [41] in which the different CRDs of Gal-9 mediated opposing functions in a tube formation assay. The C-terminus of Gal-9 suppresses endothelial sprouting; conversely, the N-terminus promotes endothelial sprouting. In agreement with other reports, Rao et al. showed that treating cells with Gal-4 antibody increased cell proliferation and treating cells with recombinant Gal-4 increased cell cycle arrest, causing apoptosis through p27 induction and suppressing cyclin D1 and c-Myc expression in colorectal cancer [42]. These diverse findings indicate the need to better understand the functionality of different Gals in different cell types. Several studies have demonstrated that Gals regulate the growth of cancer cells. Gal-1 has been reported to participate in immune surveillance escape to regulate colorectal cancer growth [43]. Another study from Liang and coworkers found that Gal-3 enhances tumor initiation in hepatocellular carcinoma [44]. Furthermore, Liebscher et al. observed that Gal-1 regulates neuroblastoma cell growth, and knockdown of Gal-1 induced cell apoptosis [45]. Gal-9 promotes cell proliferation and was negatively regulated by lncMX1–215-mediated H3K27 acetylation on the promoter, and treating cells with histone deacetylase inhibitors increased metastasis in head and neck squamous cell carcinoma [46]. Similar observations were described by Enninga et al. who found that Gal-9 induces tumor growth by regulating CD206 macrophages in melanoma [47]. Furthermore, many studies have confirmed that the expression of Gals is associated with cancer stemness and resistance to drug treatment. Gal-1 increases invasion by stabilizing Ras to control the ERK pathway and promotes castration-resistant prostate cancer progression [48]. Cristiani et al. showed that the co-expression of PD-L1 and galectin-9 increases lung cancer sphere formation [49]. Mechanistically, Gal-9 promotes a stemness phenotype by modulating CCR7–CCL19 axes. This effect is in agreement with another cancer type wherein Gal-3-mediated immunosuppressive was required for prostate cancer stemness and metastasis [50]. Interestingly, Gal-3 is also reportedly increased in lung cancer, and monocytic MDSC with Gal-3 may induce cellular resistance to chemotherapy treatment [51], confirming that posttranslational modification of Gal affects subcellular location and functionality. 

Gal also participates in other processes during cancer progression, including angiogenesis and motility. Gal-1 is reportedly induced by communicating glycosylated receptors to regulate cell angiogenesis [52]. In colon and breast cancer, Gal-3 can interact with glycoprotein VI, and activation of Gal-3 from cancer cells promotes extravasation by stimulating activation and degranulation in platelets [53]. Similar observations were described for in vivo experiments; endogenous Gal-8 can be secreted by MCF-7 cells, which increases microvascular permeability by binding to integrin-β1 and VEGFR2 to activate AKT-eNOS-FAK signaling that promotes angiogenesis and metastasis [54]. Conversely, in multiple myeloma patients, knockdown of Gal-1 upregulates MMP9, CCL2, SEMA3A, and CXCL10 to promote angiogenesis [55]. Additionally, Wu et al. and coworkers showed the interaction of Gal-3 and secreted carcinoembryonic antigen in colorectal cancer cells; knockdown of Gal-3 blocked carcinoembryonic antigen-mediated cell migration and metastasis [56]. Taken together, these findings indicate that molecules associate with Gal to affect the functions of Gal. The functions of these molecules are also affected by the presence or absence of Gal. As described above, several factors influence Gal functions, including posttranslational modification of mRNA or protein, subcellular location, and interacting partners. According to current studies, the mechanism by which Gal is secreted into the extracellular environment remains unknown because of its lack of a secretion signal peptide or transmembrane domain [57]. For example, posttranslational modification or interaction partners may shift Gal’s location. Gong et al. showed that phosphorylation of Gal-3 on the N-terminal domain alters its subcellular location, particularly residues 89-96 [7,58]. Phosphorylation of Gal-3 at tyrosine by calpain 4 has been found to increase extracellular secretion [59]. Interestingly, artificial plus acylation sequences accelerate Gal secretion in cos-7 cells [60]. Sato and coworkers found that Gal can regulate baby hamster kidney cell attachment and spreading, and the secretion of Gal is affected by methylamine, serum starvation, heat shock, and calcium ionophores [61]. Mutation of arginine to alanine at residue 224 was critical for nuclear localization of Gal-3 and protein stability [62]. Furthermore, inhibition of miR-1275 by RACK1 induced Gal-1 expression and secretion in cervical cancer [63]. Gal-3-binding proteins, such as synexin and importin, were found to be associated with intracellular Gal-3 and translocation into the nucleus [62,64,65], revealing that identifying Gal-binding proteins may unveil mechanisms of Gal translocation. In addition to these molecules, scientists have also found various potential interaction partners with the Gal family. Thus, these binding partners of Gal have been collected on the website through prediction or experimental proof (Table 4). 

Moreover, many studies have confirmed that Gals interact with different partners or have different functions in different locations. Thus, distinguishing among Gal locations (extracellular and intracellular) may unveil these roles. Table 5 is organized based on the current observations. However, it is worth noting that Gal’s expression in subcellular localization is regulated in response to different types of stimuli. Using the widely discussed Gal-3 as an example, its expression is stimulated by growth factors, cytokines, environmental changes, death signals, and in response to drug treatment [101,102,103,104]. Through stimulus-induced activation of intracellular signaling, such as Ras/MAPK/ERK, Smad signaling increases the binding of transcription factors to the Gal-3 promoter [101,105]. Park et al. demonstrated that Toll-like receptor 4 activates Gal-1 expression by stimulating lipopolysaccharide in colon cancer cells [79]. Interestingly, as an autocrine molecule, Gal can regulate its own expression by associating with extracellular receptors, such as integrin, EGFR, VEGFR2, and BMPR (Figure 3) [101,105]. For example, Gal-1 regulates triple-negative breast cancer progression and drug resistance by interacting with integrin-β1 to activate the integrin-β1/FAK/c-Src/ERK/STAT3/survivin pathway [72]. Recently, Oysnsdel et al. showed that Gal-8 interacts with integrin α5β1 to induce epithelial transformation into a mesenchymal-like phenotype [106]. The interaction of Gal-3 and EGFR may partially mediate MUC1 to promote cancer progression [107]. Importantly, this mechanism may increase the crosstalk between different signaling pathways and enable cells to respond to different stimuli. Seguin demonstrated that the interaction of Gal-3 and integrin-β3 bypasses the inhibition of EGFR inhibitors, promoting drug resistance and stemness [108]. Conversely, Gal-3 is a negative regulator of melanoma that acts by regulating integrin-β3 expression [109]. EGFR inhibitor treatment causes increased integrin αvβ3 expression, resulting in drug resistance by activating Gal-3/KRAS/RalB/TBK1/NF-κB signaling in non-small cell lung cancer [91]. However, how Gal is transactivated by unique mechanisms in response to drug treatment is still unknown.

## 5. The Roles of Galectins in Cancer Metabolism Reprogramming

Common metabolic abnormalities involve sugars, lipids, proteins, and nucleotides. Abnormal metabolism leads to long-term inflammation in cells and promotion of tumorigenesis [110]. Cancer cells can also improve cell survival by changing their metabolism and adapting the way nutrients are taken up [111]. Interestingly, cancer cells express increased glucose transporter (GLUT) expression levels to obtain increased nutrients [112]. Understanding tumorigenesis caused by metabolic abnormalities and how cancer cells grow in response to metabolic abnormalities has become the subject of multiple studies [110,111,112]. Moreover, metabolic disorders are usually accompanied by organ fibrosis, such as diabetes mellitus-mediated aortic stenosis or lung fibrosis [15,37]. Evidence shows that patients with pulmonary fibrosis may be more prone to lung cancer development [113,114,115,116]. Tang et al. revealed that Gal-1 was overexpressed in pancreatic stellate cells that participated in chronic pancreatitis and pancreatic cancer progression [96]. However, cancer patients also exhibit fibrosis and increased Gal-1 production during progression or chemotherapy [96,117]. Radiation-induced pulmonary fibrosis was observed to increase Gal-1 expression in a murine model [118]. Thus, these findings indicate that both the abnormal expression of Gal and cancer therapy result in fibrosis (Figure 4). Additionally, metabolic reprogramming is considered one of the hallmarks of fibrosis [119]. Gals have abnormal expression patterns in metabolic abnormalities, and changes in Gal have been identified in the progression of many cancers [120].

### 5.1. Carbohydrate Metabolism

Zheng et al. reported that Gal-1 expression was correlated with tumor volume and glycolysis-related markers (GLUT-1 and hexokinase II), which may serve as an independent prognostic marker in lung adenocarcinoma [89]. Park et al. showed that Toll-like receptor 4 (TLR4) increases Gal-1-mediated ADAM metallopeptidase domain 10 (ADAM10) and ADAM17, which promotes lactate (2-hydroxypropanoic acid) production [79]. Lactate is the primary metabolite of glucose through anaerobic glycolysis in normal cells [121]. Lactate was once considered a waste product in the metabolic process [121]. In some tissues, such as the liver, brain, heart, and skeletal muscle, lactate can serve as a source of energy or as a carbon source for gluconeogenesis through the Cori cycle in the liver [122]. In addition, cancer cells can generate energy through the Warburg effect to accelerate ATP production [123]. However, anaerobic glycolysis produces approximately 85% lactic acid that is transported to the extracellular environment and acidifies the microenvironment [124]. Furthermore, Apicella et al. showed that tumor-associated stromal cells, such as cancer-associated fibroblasts, could take up lactate to stimulate production and to increase tumor cell resistance to therapy [125], indicating that Gal promotes lactate production and may create a more acidic microenvironment for tumor cells. In addition, cancer cell-mediated lactate production triggers hypoxia-inducible factor 1α (HIF-1α) expression under hypoxic conditions [126] and the expression of glucose transporters but also the regulated expression of Gal-1 [89], which may form a regulatory loop between tumor cells and tumor-associated cells. Gal-1 is highly expressed in lymphoma, and its concentration was correlated with lactate dehydrogenase (LDH) expression [127]. Moreover, more aggressive metastatic cells exhibit high Gal-1 expression and LDH B expression in melanoma [128].

### 5.2. Amino Acid Metabolism

In contrast, glutamine, which is an energy source, can be incorporated into the tricarboxylic acid (TCA) cycle, and while glutamine is a primary energy resource, tumor cells can provide more lactate [129]. However, Li et al. confirmed that glutamine synthesis and uptake can be negatively regulated by Gal expression, and showed that upregulation of Gal-1 was significantly associated with reduced glutamine synthetase expression in urinary bladder urothelial carcinoma [66]. Similar results can be observed when recombinant Gal-4 treatment inhibits tumor growth by decreasing the phosphorylation of SLC1A5 (a glutamine transporter) in colon cancer cells [83], and Kazenmaier et al. identified that Gal-12 can bind to SLC1A5 to reduce glutamine uptake in colon cancer cells [81].

### 5.3. Lipid Metabolism

Fatty acids are an essential energy source for cancer metabolism as well. In healthy cells, obesity caused by an abnormal diet can reduce weight gain by targeting Gal-1 [130]. In tumor cells, Mukherjee et al. observed that Gal-12 regulates lipid draft formation to inhibit human promyelocytic leukemia lipogenesis [69].

Energy generation and metabolism primarily come from the mitochondria. Consequently, these findings show that overall mitochondrial metabolism is also altered during the process of carcinogenesis (Figure 5) in which the TCA cycle is central for the conversion of different metabolites to produce lactate for reprogramming the tumor microenvironment.

### 5.4. Disorder of Mitochondria

Mitochondria also regulate cell survival through the intrinsic pathway, and abnormal mitochondrial function was observed in the absence of Gal-3 [131]. Yu et al. showed that Gal-3 regulates cytochrome c release to prevent mitochondrial damage in human breast epithelial cells [64]. Wang et al. found that cisplatin-induced mitochondrial dysfunction is inhibited by Gal-3 in ovarian carcinoma [93]. Similarly, targeting of Gal-3 decreased bcl-2 protein levels in ovarian carcinoma [94]. Tadokoro et al. reported that treating cells with Gal-9 induced mitochondrial release of apoptosis-related molecules, such as cytochrome c, Smac/Diablo, and HtrA2, inhibiting liver cancer proliferation [88]. In agreement with other reports, Chiyo and coworkers demonstrated that Gal-9 induces apoptosis in esophageal squamous cell carcinoma through the mitochondria [85]. Sakhnevych et al. found that Gal-9 and its receptor Tim-3 form a complex and accumulate in the mitochondria in response to a Bcl-X_L_ antagonist in colorectal cancer cells [84]. Extracellular Gal-7 can reenter cells and translocate into the nucleus to interact with bcl-2 as an anti-apoptosis function to promote breast cancer chemoresistance [132]. Altogether, these findings indicate that Gal participates in metabolic reprogramming and mitochondrial dysfunction during cancer progression. Dysregulated metabolism-mediated Gal expression contributes further to microenvironment alterations.

## 6. Galectin in the Microenvironment

The interactions in the tumor microenvironment are considered another critical mechanism for cancer progression that assimilate peripheral cells through secretion of various molecules [133,134]. Furthermore, there are various cells surrounding the microenvironment, such as stromal cells and immune cells, which are reportedly regulated by Gals [82,135,136]. Additionally, immune evasion is one of the most important mechanisms for cancer survival [137]. The interactions of extracellular Gal not only activate downstream cell signaling pathways but also contribute to environmental reprogramming (Figure 6). For example, increased Gal-3 levels are often detected in cancer patients’ blood, indicating that circulating Gal contributes to microenvironment reprogramming during cancer progression [75,138]. Circulating Gal-3 interacts with endothelial cells, which induce cytokine production that promotes cancer progression [139,140]. Colomb et al. showed that Gal-3 interacts with CD146 on endothelial cells through affinity purification assays, which leads to AKT signaling activation and IL-6 and G-CSF secretion to promote cancer progression [141]. Croci and coworkers demonstrated that tumors develop resistance to anti-VEGF therapy by secreting Gal-1 to interact with VEGFR2 in endothelial cells [142]. Tumor-mediated Gal-1 and Gal-3 have also been identified as inhibiting T cell cytotoxicity by interacting with the T cell receptor or lymphocyte activation gene 3 (LAG-3) and inducing T cell apoptosis [143,144]. In addition, Gal-3 functions as a switch for macrophage polarization and regulates CD8^+^ T cell infiltration into lung adenocarcinoma cells [92]. Treatment with Gal-3 antagonists promotes T cell infiltration by recognizing cancer-mediated interferon-gamma in vivo [145]. It has been reported that the expression of Gal-1 is a marker of lymphocyte infiltrates in cutaneous head and neck cancers [146] and triple-negative breast cancer patients [73]. Evidence also shows that prostate cancer-mediated Gal-1 downregulates lymphocyte proliferation and apoptosis [98]. Tesone et al. showed that tumor-associated macrophages express Gal-9 to promote cancer progression [68]. Overexpression of Gal-1 activates hepatocellular carcinoma and promotes cancer cell immune surveillance escape by inducing T cell apoptosis [147]. Andersen et al. revealed that circulating Gal-1 in serum might promote M2 macrophage activation in multiple myeloma patients [70]. Suppression of Gal-1 in glioma cells sensitized them to natural killer cells (NK cells), which was caused by cancer cells producing more pro-inflammatory cytokines for recruitment of monocytic myeloid-derived suppressor cells to differentiate into dendritic cells, leading to further recruitment of NK cells [148,149]. Moreover, treating melanoma patients with a BRAP/MEK inhibitor increased Gal-1 expression by an unknown mechanism that may lead to immune surveillance escape and cause drug resistance **[99]**. Similarly, Gal-9 has been found to be associated with immune tolerance during pregnancy [150]. In contrast, tumor-associated macrophages expressing Gal-9 are associated with invasive bladder tumor stage and decreased immune surveillance **[68]**. In addition, evidence suggests that Gal-9 binds to CD206 on M2 macrophages to induce the secretion of angiogenesis factors to promote tumor growth **[47]**. Additionally, evidence has revealed that Gal-9 interacts with its receptor Tim3 on T cells, prompting immunosuppression of the tumor microenvironment **[151,152]**. However, Luo et al. demonstrated that treating T cells with Gal-7 suppresses TGF-β signaling to activate polarization toward CD4 T cells **[153]**. In contrast, Higareda-Almaraz and coworkers showed that Gal-7 is a negative regulator of cervical cancer that acts through reprogramming the tumor microenvironment [78]. Interestingly, fibroblasts were shown to be activated by cancer cells known as cancer-associated fibroblasts, and the expression of Gals was associated with fibroblast activation and promoted growth [135,136]. Toti et al. found that knockdown of Gal-1 in tumor stromal cells, such as human pancreatic stellate cells, decreased pancreatic ductal adenocarcinoma growth in vivo [97]. In addition, cancer-associated fibroblasts mediate Gal-1 regulation of cancer cell motility through macrovesicle release, and knockdown of Gal-1 prevents cancer-associated fibroblast-mediated prostate and pancreatic cancer migration [136]. This finding is in agreement with previously described cancer-associated fibroblasts expressing Gal-1 to promote melanoma cell migration [154]. The possible mechanism might be due to Gal-1’s interaction with integrin-β1 on fibroblasts to activate Gli1 expression, resulting in increased metastasis of gastric cancer [86]. In addition, Gal-1 secreted by cancer cells can induce cancer-associated fibroblasts to activate TDO2-ATK signaling and produce the tryptophan metabolite kynurenine [90], which reportedly can induce T cell apoptosis [155,156]. Furthermore, bone remodeling is an essential mechanism for cancer cell interactions with bone marrow stromal cells in the microenvironment. Gals have been reported to participate in cancer-mediated bone remodeling [71,76,157]. Muller and coworkers found that Gal-1 expression was important for multiple myeloma development and bone mass [71]. Evidence shows that Gal-3 cleavage is observed in metastatic bone cancers and plays a different role than Gal-3 during osteoclastogenesis, and cancer-mediated Gal-3 regulates osteoclastogenesis by binding to myosin-2A in breast and prostate cancer [76]. A murine model revealed that depletion of Gal-3 increases bone metastasis in breast cancer through the CXCR4/CXCL12 axis [157].

## 7. Available Inhibitors for Targeting Galectins

Based on the reported Gal functions in disease progression above, investigators have attempted to design specific inhibitors to block Gals’ function as therapeutic reagents for related diseases. There are various methods that have been used to block Gal functions, such as neutralizing antibodies, synthetic compounds, carbohydrate derivatives, and binding peptides. Several inhibitors used in clinical trials for testing in multiple human diseases in different phases are listed in Table 6.

### 7.1. Neutralizing Antibodies

Neutralizing antibodies are commonly used to block protein functions. Stasenko et al. showed that invasion and proliferation were suppressed in high-grade serous ovarian cancer, and showed improved overall survival, in response to treatment with an anti-Gal-3 antibody [158]. Evidence shows that neutralizing antibodies may partially suppress the inhibition of lactose treatment [159]. In addition, an in vivo model showed that ischemia-induced angiogenesis was decreased by treatment with neutralizing anti-Gal-3 [160]. Saez et al. demonstrated that abnormal angiogenesis can be blocked by a specific Gal-1 neutralizing antibody [52]. Using a neutralizing antibody to block the expression of Gal-9 in colon adenocarcinoma resulted in increased T cell cytotoxicity and immunosuppression [161]. Metastatic melanoma adhesion to endothelial cells can be blocked by response to Gal-1 antibody [162]. Moreover, β4-integrin/PI3K activation to epidermoid carcinoma cells can be blocked by treating cells with a Gal-3 antibody [104]. In vivo, targeting Gal-3 with antibodies inhibited B16 melanoma and UV-2237 fibrosarcoma metastasis to the lung [163]. Similarly, Nakamura et al. found that an anti-Gal-3 antibody inhibited adhesion and liver metastasis of adenocarcinoma [164]. Interestingly, drug delivery can be conducted using Gal as a target. Ma and coworkers found that anti-Gal-3-based nanoparticles could control drug delivery and increase the concentration of doxorubicin in thyroid cancer in vivo [165]. In contrast, blocking Gal-4 using a neutralizing antibody on the cell surface induced cell proliferation [42]. Blocking Gal-3 also boosted cytokine INF-gamma secretion of CD8(+) tumor-infiltrating lymphocytes [143,166]. Dovizio et al. found that anti-Gal-3 antibodies prevented cox-2 expression during platelet adhesion in colon carcinoma [80]. Gal-3 neutralizing antibody also inhibited Gal-3-mediated ERK signaling as well as neuroblastoma-mediated IL-6 expression in bone marrow stromal cells [167]. Targeting Gal-3 with an antibody blocked tumor adhesion to endothelial cells, inhibiting cancer metastasis [168]. Furthermore, targeting Gal interaction partners can also block Gal functions. For example, blocking CD166, a Gal-8 binding partner, suppresses Gal-8-mediated migration and tube formation of endothelial cells [169].

### 7.2. Carbohydrate Derivatives

Gal interacts with its partners by recognizing N-glycans, such as N-acetyl-D-lactosamine, by the carbohydrate recognition domain; therefore, there are various ways to block this interaction. Pretreating cells with lactose blocks Gal-3-mediated cell adhesion [100]. Lactose blocks the association between Tim-3 and Gal-9, inducing immune molecule expression of T cell immunoreceptors with Ig and ITIM domains [170]. Pan et al. showed that Gal-3-mediated neutrophil infiltration could be blocked by lactose treatment to reverse acute pancreatitis [171]. Moreover, Gal antagonists, such as LacNAc (N-acetyl-D-lactosamine) and TetraLacNAc (tetra-N-acetyl-D-lactosamine), reduce tumor growth by inducing IFN-gamma and chemokine production to induce CD8(+) T cell infiltration [145]. Lactose treatment blocks exosomes derived from HIV-infected dendritic cells [172]. In addition, evidence shows that synthetic inhibitors, such as synthetic lactulose amines, have been reported to suppress tumor progression by binding to Gal-1 and Gal-3 [159].

### 7.3. Galectin-Binding Peptides

The truncated Gal-3 carbohydrate recognition domain (Gal-3C), a Gal-3-binding peptide designed based on Gal-3′s structure, was found to inhibit cancer progression, which was generated by matrilysin-1 [173,174]. John et al. showed that Gal-3C suppresses breast cancer growth and metastasis in vivo [77]. This outcome is in agreement with previously described Gal-3C inhibiting multiple myeloma development and synergistically boosting the effect of bortezomib [175]. Evidence also shows that recombinant Gal-3C regulates the integrin/FAK/SRC/NDRG1 axis to suppress hepatocellular carcinoma progression [176]. Identifying peptides that can interfere with the carbohydrate recognition domain of specific Gals, such as Thomsen–Friedenreich antigen-specific peptide P-30, may disrupt Gal functions [177,178]. Lian and coworkers identified the Gal-binding peptide G3-C12-HPMA-KLA, which has dual effects on cancer cells and subcellular mitochondria [179]. In addition, the Gal-1-binding peptide “anginex” was designed to induce apoptosis of endothelial cells and to decrease angiogenesis [180]. Anginex-conjugated arsenic–cisplatin combined liposomes enhanced therapeutic efficiency [74]; evidence suggests that anginex may also have dual activity by suppressing H-Ras translocation [181].

### 7.4. Synthetic Compounds

Non-peptide compounds, such as OTX008, have been designed to target Gal-1 based on the effect of anginex [87], which has been found to serve as a therapeutic reagent for cancer and diabetic patients [40,87,182,183,184]. Treatment with BH3I-1 blocks the interaction between Tim-3 and Gal-9, which increases immune surveillance in colorectal cancer [84]. In addition, plant-derived galactomannans (known as DAVANAT) have been found to bind both Gal-1 and Gal-3, leading to CD8(+) tumor-infiltrating lymphocyte cytotoxicity [166,185,186]. The same effect can be observed from another Gal-3 inhibitor, “GR-MD-02” (also known as belapectin), which has been identified to reduce liver fibrosis [187,188]. Saccharide derivatives have been found to inhibit Gal functions and include related inhibitors, such as GCS-100, a modified form of citrus pectin (MCP), RN1, and disaccharide thiodigalactoside (TDG) [189]. Chauhan et al. showed that GCS-100 activates NF-KB signaling to induce apoptosis in lymphoma [189]. In agreement with other reports, the authors also showed that GCS-100, an antagonist of Gal-3, induced cancer cell death [190,191,192], boosted tumor-infiltrating T lymphocyte secretion of IFN-gamma [143], and increased prostate cancer sensitivity to cisplatin [193]. Similarly, an effect was observed in response to modified citrus pectin (MCP) on cancer [67,95,194,195]; however, MCP can also be used for other Gal-related disease treatments, such as decreasing Gal-3 levels in type 2 diabetes [196], decreasing doxorubicin-induced cardiovascular diseases [197], ameliorating cardiac dysfunction [198], and improving ischemic heart failure [199]. In addition, the calixarene derivative compound “OTX-008” was designed as a Gal-1 inhibitor [200]. Evidence shows that OTX-008 suppresses cancer progression by targeting Gal-1 [87,182,184,200]. Moreover, OTX-008 treatment blocks Gal-1-mediated retinal neovascularization, renal fibrosis in diabetes, and proliferative diabetic retinopathy [40,183,201]. The Gal-3 inhibitor “GB1107” also increased PD-L1-mediated immune surveillance in lung adenocarcinoma [92].

### 7.5. Other Derivatives

Other polysaccharide derivatives include “RN1” and “HH1-1.” Zhang and coworkers demonstrated that RN1 binds to Gal-3 and blocks Gal-3-mediated downstream signaling, which suppresses transcription factors, such as RUNX binding to the Gal-3 promoter in pancreatic ductal adenocarcinoma [105]. Yao et al. showed that HH1-1 blocks the interaction between Gal-3 and EGFR, which decreases EGFR/AKT/FOXO3 signaling to halt the progression of pancreatic cancer [202]. The same effect can be observed with another Gal-1 inhibitor, thiodigalactoside (TDG) [203]. Lai et al. found that sorafenib-resistant breast cancer-induced Gal-1 was suppressed by TDG treatment [204]. In agreement with other reports [205,206], the authors also showed that TDG increased both CD4 (+) and CD8 (+) T cells by blocking Gal-1 in murine breast, colon, and lung cancer models. TDG halted Gal-1-induced cisplatin in hepatocellular carcinoma [207]. Moreover, targeting Gal-1 through TDG treatment suppressed diet-induced obesity [208]. Similarly, this effect can be observed in “TD139,” a specific Gal-3 inhibitor derived from thiodigalactoside, which blocks TGF-β-induced lung fibrosis [209]. In addition, several Gal-1-related inhibitors, such as lactobionic acid (LBA), inhibit diet-mediated obesity [210], and act as a third-generation photosensitizer (PS) that increases the cytotoxicity of irradiation in bladder cancer [211]. Moreover, LLS2 treatment increases paclitaxel-induced cytotoxicity in ovarian cancer [212].

## 8. Conclusions

Homeostasis of the metabolic process is vital for cell growth and stability. Abnormal metabolism promotes a long-term inflammatory response, which also contributes to fibrosis and accompanying tumorigenesis. The Gal family is also involved in these processes. Gal exists in different tissues and cells. According to current research on Gal, the functions of Gal are very complicated, which may be due to its posttranscriptional modifications, subcellular localization, cell type, and interacting partners. In particular, several studies have indicated that the functions of extracellular Gal are different from those of intracellular Gal [132,213]. This may be due to the extracellular form of Gal being modified by other mediators or interacting with different molecules that promote distinct functions. Gal has also been found to interact with different molecules through non-carbohydrate binding to regulate different cellular functions [8,9]. Therefore, it is imperative to analyze and to identify molecules and functions inherent to Gal. However, the current understanding shows that Gal interactions and distribution are not thoroughly understood. It is essential to distinguish regulators of Gal expression in response to environmental pressures. For example, when cells receive chemotherapy drugs long term, Gal may be induced by drug resistance, resulting in more drug resistance, and hindering clinical treatment. Therefore, extensive studies are needed to fully understand Gal. According to Table 5, different Gals can exist in the same subcellular compartments; however, whether they function synergistically or regulate one another is still uncertain. In particular, extracellular Gal has also been found to be a mediator that affects cell–cell interactions. Based on current evidence, we know that when abnormal cell metabolism or tumorigenesis occurs, certain Gals are upregulated to promote organ fibrosis. However, dysregulation of metabolic disease-induced fibrosis promotes cancer progression by inducing Gal in the tumor microenvironment, but precisely how this occurs is still unclear. Therefore, systematically investigating how these interactions are regulated by Gal in the microenvironment and the complex molecular mechanisms involved may further enhance the effects and reduce the toxicity caused by non-specificity in the design of specific inhibitory drug treatments. Whether inhibition will achieve therapeutic effects needs further research and discussion. Furthermore, designing inhibitors of partners that associate with Gal and combinations with current therapeutic drugs may achieve a synergistic response in Gal-related disease.

## Figures and Tables

**Figure 1 biomedicines-09-01159-f001:**
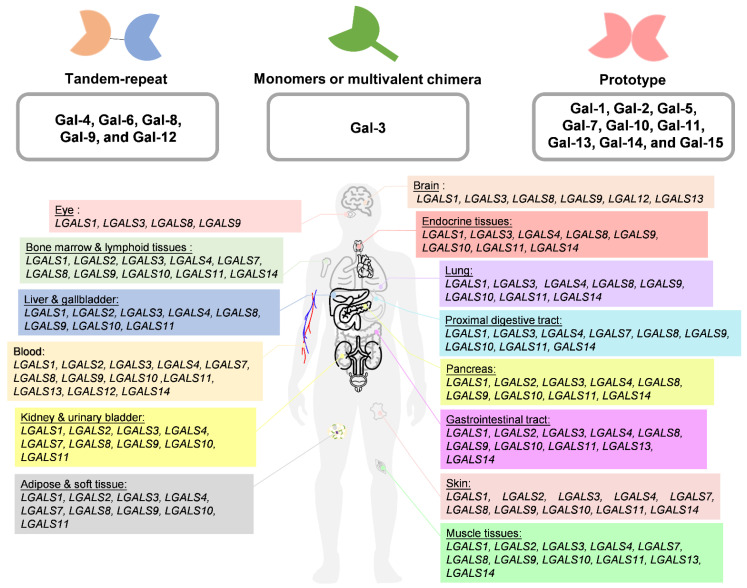
The groups of Gal’s family structures. The distribution of galectin (*LGALS*) is dependent on the RNA level in the human body. All profiles were collected from the human protein atlas website (https://www.proteinatlas.org/, accessed on 15 September 2020) (see Appendix A).

**Figure 2 biomedicines-09-01159-f002:**
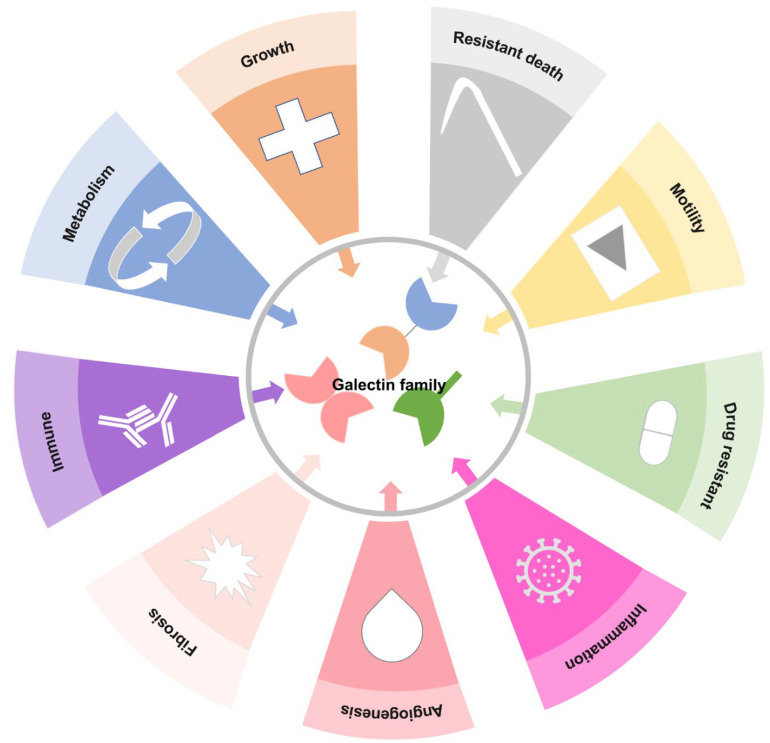
Pluripotency of the galectin family in hallmarks of cell biology. The schematic diagram illustrates previous Gal functions; Gals are involved in cell proliferation, migration, anti-apoptosis, carbohydrates, proteins, lipid metabolism or nucleic acid synthesis, resistance to drug treatment, immune system or inflammation response, vessel generation, organ fibrogenesis, and resistance to death.

**Figure 3 biomedicines-09-01159-f003:**
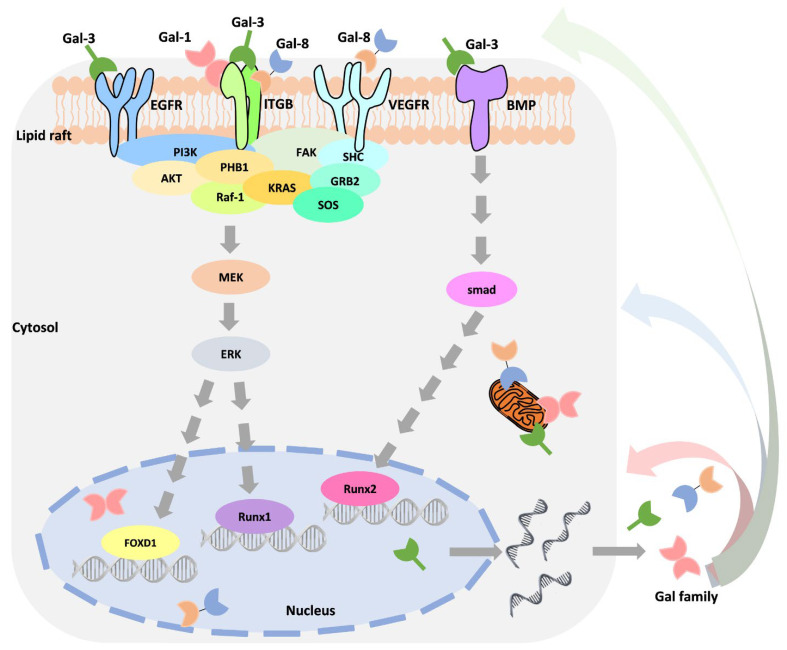
Crosstalk of various signaling molecules that regulate galectin expression. The schematic diagram illustrates the complex regulation involved in Gal expression and interaction components.

**Figure 4 biomedicines-09-01159-f004:**
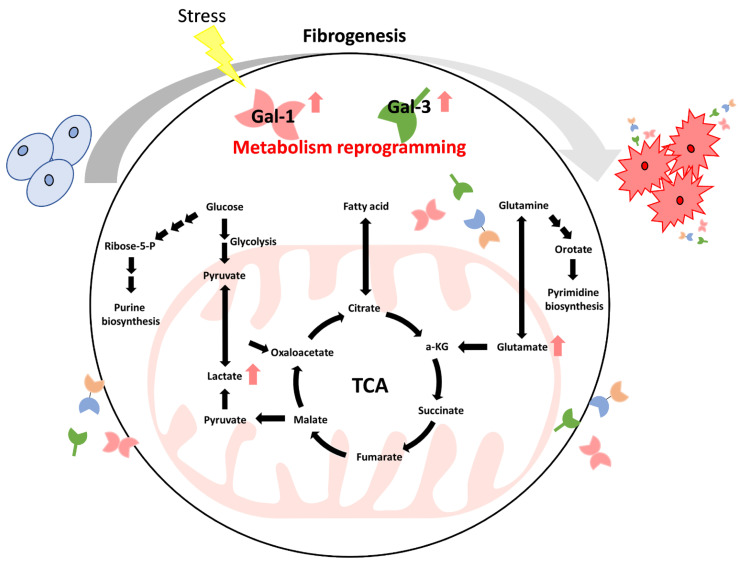
Metabolic reprogramming-associated fibrogenesis induction of galectin expression. The schematic diagram illustrates that metabolic reprogramming accompanies organ fibrogenesis and dysregulated Gal expression during stress stimulation.

**Figure 5 biomedicines-09-01159-f005:**
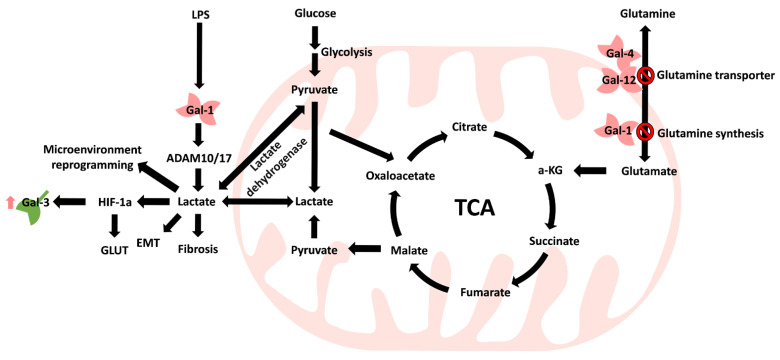
Galectin family involvement in mitochondrial metabolism reprogramming. Gal expression accompanies both metabolism and microenvironment reprogramming, which can be recognized in several diseases, including tumor progression. ↑ Means increase.

**Figure 6 biomedicines-09-01159-f006:**
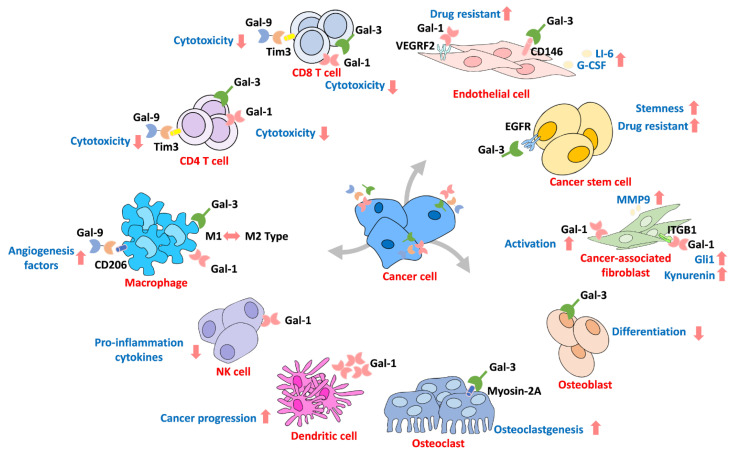
The galectin family contributes to tumor microenvironment reprogramming. Gals can act as a reprogramming messenger to connect tumor cells and other cells in the microenvironment. ↑, ↓ Means increase or decrease.

**Table 1 biomedicines-09-01159-t001:** Relative RNA expression of galectin (*LGALS*) in organs.

	*LGALS* Expression										
All Organs	*LGALS1*	*LGALS2*	*LGALS3*	*LGALS4*	*LGALS7*	*LGALS8*	*LGALS9*	*LGALS10*	*LGALS12*	*LGALS13*	*LGALS14*
Brain	●		●			●	●		●	●	
Eye	●		●			●	●				
Endocrine tissues	●		●	●		●	●	●			●
Lung	●		●	●		●	●	●			●
Proximal digestive tract	●		●	●	●	●	●	●			●
Gastrointestinal tract	●	●	●	●		●	●	●		●	●
Liver and gallbladder	●	●	●	●		●	●	●			
Pancreas	●	●	●	●		●	●	●			●
Kidney and urinary bladder	●	●	●	●	●	●	●	●			
Muscle tissues	●	●	●	●	●	●	●	●		●	●
Adipose and soft tissue	●	●	●	●		●	●	●			
Skin	●	●	●	●	●	●	●	●			●
Bone marrow and lymphoid tissues	●	●	●	●	●	●	●	●			●
Blood	●	●	●	●	●	●	●	●	●	●	●

● Means positive expressed. All profiles were collected from the human protein atlas website (https://www.proteinatlas.org/, accessed on 15 September 2020).

**Table 2 biomedicines-09-01159-t002:** The prognostic role of galectin family members of pan-cancer in TCGA cohorts.

				Prototype																	
Cancer Type	Endpoint	Case	Cohort	*LGALS1*	HR	Risk	*LGALS2*	HR	Risk	*LGALS7*	HR	Risk	*LGALS10*	HR	Risk	*LGALS13*	HR	Risk	*LGALS14*	HR	Risk
ACC	Overall survival	76	TCGA	0.7800	1.1	high	0.2300	0.7	low	N.S.			0.7600	0.9	low	N.S.			N.S.		
BLCA	Overall survival	402	TCGA	0.0450	1.4	high	0.3800	0.9	low	0.1100	1.3	high	0.9200	1.0	low	N.S.			N.S.		
BRCA	Overall survival	1070	TCGA	0.1600	1.3	high	0.3300	0.8	low	0.5400	1.1	low	N.S.			N.S.			N.S.		
CESC	Overall survival	292	TCGA	0.3100	1.3	high	0.1400	0.6	low	0.0510	0.6	low	0.0089	0.4	low	N.S.			N.S.		
CHOL	Overall survival	36	TCGA	0.0980	2.2	high	0.1700	0.5	low	0.5500	1.3	high	N.S.			N.S.			N.S.		
COAD	Overall survival	270	TCGA	0.2700	1.3	high	0.8400	1.0	low	0.9100	1.0	high	0.1500	0.7	low	N.S.			N.S.		
DLBC	Overall survival	46	TCGA	0.6200	0.7	low	0.1200	0.4	low	N.S.			0.5300	0.7	low	N.S.			N.S.		
ESCA	Overall survival	182	TCGA	0.6300	1.1	high	0.1300	0.7	low	0.8900	1.0	low	0.9900	1.0	high	N.S.			N.S.		
GBM	Overall survival	161	TCGA	0.4500	1.2	high	0.3300	1.2	high	0.3800	1.2	high	1.0000	1.0	low	N.S.			N.S.		
HNSC	Overall survival	518	TCGA	0.0420	1.4	high	0.9900	1.0	high	0.1700	1.3	high	0.1700	0.8	low	N.S.			N.S.		
KICH	Overall survival	64	TCGA	0.0250	5.0	high	0.9400	1.1	high	N.S.			N.S.			N.S.			N.S.		
KIRC	Overall survival	516	TCGA	0.0023	1.8	high	0.0003	0.5	low	0.6400	0.9	low	0.0058	0.6	low	N.S.			N.S.		
KIRP	Overall survival	282	TCGA	0.3100	1.3	high	0.1800	1.5	high	0.6100	0.9	low	N.S.			N.S.			N.S.		
LAML	Overall survival	106	TCGA	1.0000	1.0	N.S.	1.0000	1.0	N.S.	N.S.			1.0000	1.0	N.S.	N.S.			N.S.		
LGG	Overall survival	514	TCGA	0.0061	1.5	high	0.0062	0.7	low	N.S.			N.S.			N.S.			N.S.		
LIHC	Overall survival	364	TCGA	0.4800	1.1	high	0.0230	0.7	low	N.S.			N.S.			N.S.			N.S.		
LUAD	Overall survival	478	TCGA	0.4500	1.1	high	0.0230	0.7	low	0.5600	1.1	high	0.0370	0.7	low	N.S.			N.S.		
LUSC	Overall survival	482	TCGA	0.0470	1.4	high	0.4600	1.1	high	0.5600	0.9	low	0.9600	1.0	low	N.S.			N.S.		
MESO	Overall survival	82	TCGA	0.2200	1.4	high	0.6200	0.9	low	N.S.			N.S.			N.S.			N.S.		
OV	Overall survival	424	TCGA	0.1500	1.2	high	0.1500	0.8	low	0.7800	1.0	N.S.	0.3300	0.9	low	N.S.			0.9900	1.0	high
PAAD	Overall survival	178	TCGA	0.0680	1.5	high	0.3600	0.8	low	0.4500	0.9	low	0.0059	1.9	high	N.S.			N.S.		
PCPG	Overall survival	182	TCGA	0.7400	1.2	low	0.3800	0.7	low	N.S.			N.S.			N.S.			N.S.		
PRAD	Overall survival	492	TCGA	0.7900	1.1	high	0.2900	1.3	high	0.0780	0.7	low	N.S.			N.S.			N.S.		
READ	Overall survival	92	TCGA	0.1900	1.8	high	0.6500	1.2	high	0.2900	1.8	high	0.3100	0.6	low	N.S.			N.S.		
SARC	Overall survival	262	TCGA	0.2500	1.2	high	0.1000	0.7	low	N.S.			N.S.			N.S.			N.S.		
SKCM	Overall survival	584	TCGA	0.2100	1.2	high	0.1900	0.9	low	0.0270	1.3	high	N.S.			N.S.			N.S.		
STAD	Overall survival	384	TCGA	0.4500	1.2	low	0.2500	0.8	low	0.0077	1.7	high	0.2200	1.3	high	N.S.			N.S.		
TGCT	Overall survival	136	TCGA	0.0970	1.9	high	0.2900	1.5	high	0.8900	1.1	high	0.5800	1.2	high	N.S.			N.S.		
THCA	Overall survival	450	TCGA	0.7900	0.9	N.S.	0.2300	1.4	high	0.7000	1.1	high	N.S.			N.S.			N.S.		
THYM	Overall survival	118	TCGA	0.0800	0.4	low	0.7700	1.1	high	0.6100	1.3	low	0.0310	3.2	high	N.S.			N.S.		
UCEC	Overall survival	172	TCGA	0.2200	0.7	low	0.1300	0.6	low	0.7300	0.9	low	N.S.			N.S.			N.S.		
UCS	Overall survival	56	TCGA	0.6800	0.9	low	0.4900	0.8	low	0.0960	0.6	low	0.0140	0.4	low	N.S.			N.S.		
UVM	Overall survival	78	TCGA	0.0007	5.6	high	0.1500	2.0	high	N.S.			N.S.			N.S.			N.S.		
				**Tandem**												**Chimeric**					
**Cancer Type**	**Endpoint**	**Case**	**Cohort**	** *LGALS4* **	**HR**	**Risk**	** *LGALS8* **	**HR**	**Risk**	** *LGALS9* **	**HR**	**Risk**	** *LGALS12* **	**HR**	**Risk**	** *LGALS3* **	**HR**	**Risk**			
ACC	Overall survival	76	TCGA	0.0002	3.6	high	0.0001	3.8	high	0.7600	1.1	low	0.4700	0.7	low	0.0059	3.0	high			
BLCA	Overall survival	402	TCGA	0.8900	1.0	low	0.0480	1.4	high	0.0260	0.7	low	0.1900	1.2	high	0.0830	1.3	high			
BRCA	Overall survival	1070	TCGA	0.2300	0.8	low	0.2500	0.8	low	0.2700	0.8	low	0.2100	0.8	low	0.7000	1.1	high			
CESC	Overall survival	292	TCGA	0.1200	1.6	high	0.1100	1.6	high	0.0250	0.6	low	0.1500	1.5	high	0.7300	0.9	low			
CHOL	Overall survival	36	TCGA	0.4000	1.5	high	0.0830	0.5	low	0.0320	0.6	low	0.0140	0.3	low	0.5400	0.7	low			
COAD	Overall survival	270	TCGA	0.0570	0.6	low	0.3400	1.3	high	0.6100	0.9	low	0.3000	0.8	low	0.0790	0.7	low			
DLBC	Overall survival	46	TCGA	0.8800	1.1	high	0.3300	1.8	high	0.0320	7.2	high	0.5600	0.6	low	0.8300	1.2	low			
ESCA	Overall survival	182	TCGA	0.8200	1.0	high	0.4500	1.3	low	0.7900	0.9	low	0.6400	0.9	low	0.7100	1.1	high			
GBM	Overall survival	161	TCGA	0.2800	2.8	low	0.0320	1.6	high	0.2700	1.2	high	0.1400	1.3	high	0.0560	1.4	high			
HNSC	Overall survival	518	TCGA	0.6000	1.0	high	0.0390	1.4	high	0.0120	0.7	low	0.6800	0.9	high	0.2600	0.9	low			
KICH	Overall survival	64	TCGA	0.5400	1.5	high	0.4200	1.7	high	0.1200	0.3	low	0.7000	0.8	low	0.1400	3.1	high			
KIRC	Overall survival	516	TCGA	0.4000	0.9	low	0.1000	0.7	low	0.9700	1.0	low	0.0140	1.5	high	0.9500	1.0	low			
KIRP	Overall survival	282	TCGA	0.0031	2.4	high	0.2100	1.4	high	0.5500	0.8	low	0.5800	1.2	low	0.7500	1.1	high			
LAML	Overall survival	106	TCGA	1.0000	1.0	N.S.	1.0000	1.0	N.S.	0.0550	1.7	high	0.6800	1.1	high	0.8900	1.0	low			
LGG	Overall survival	514	TCGA	0.1600	1.3	high	0.0028	1.6	high	0.0004	1.9	high	0.0760	1.4	high	0.0005	1.9	high			
LIHC	Overall survival	364	TCGA	0.4800	1.1	high	0.5800	0.9	low	0.1100	1.3	high	0.3700	1.2	high	0.0037	1.7	high			
LUAD	Overall survival	478	TCGA	0.0690	0.8	low	0.3000	1.2	low	0.2500	0.8	low	0.0680	0.8	low	0.0580	1.3	high			
LUSC	Overall survival	482	TCGA	0.5700	0.9	low	0.4600	0.9	low	0.5100	1.1	high	0.6100	1.1	high	0.3600	0.9	low			
MESO	Overall survival	82	TCGA	0.6700	1.1	low	0.5200	0.8	low	0.0700	0.6	low	0.4700	1.2	high	0.9400	1.0	low			
OV	Overall survival	424	TCGA	0.8000	1.0	low	0.5800	1.1	high	0.5200	1.1	low	0.9400	1.0	high	0.6000	0.9	low			
PAAD	Overall survival	178	TCGA	0.2500	1.3	high	0.4100	1.2	high	0.0770	1.4	high	0.2400	0.8	low	0.0430	1.5	high			
PCPG	Overall survival	182	TCGA	0.8400	1.1	high	0.0060	2.6	high	0.0680	0.2	low	0.6100	0.7	low	0.3500	2.2	high			
PRAD	Overall survival	492	TCGA	0.0081	1.8	high	0.7800	0.9	low	0.2800	0.5	low	0.6100	1.4	high	0.7300	1.3	high			
READ	Overall survival	92	TCGA	0.8800	1.1	low	0.3800	1.5	low	0.5200	0.7	low	0.3600	0.7	low	0.2500	0.6	low			
SARC	Overall survival	262	TCGA	0.5700	1.1	high	0.3900	1.2	high	0.0085	0.6	low	0.9100	1.0	high	0.3100	0.8	low			
SKCM	Overall survival	584	TCGA	0.9500	1.0	high	0.3100	0.9	low	0.0004	0.6	low	0.2900	0.9	low	0.8000	1.0	high			
STAD	Overall survival	384	TCGA	0.1900	0.8	low	0.3300	1.2	high	0.2900	0.9	low	0.0047	1.6	high	0.3700	0.9	low			
TGCT	Overall survival	136	TCGA	0.8700	0.9	low	0.3800	0.7	low	0.2000	4.2	high	0.7300	0.7	low	0.0230	900,000,000.0	high			
THCA	Overall survival	450	TCGA	0.9900	1.0	high	0.5800	1.2	high	0.8700	0.9	high	0.7900	0.9	low	0.2300	0.5	low			
THYM	Overall survival	118	TCGA	0.7400	0.9	low	0.4400	0.7	low	0.9600	1.0	low	0.6300	0.7	high	0.5800	1.5	high			
UCEC	Overall survival	172	TCGA	0.8200	0.9	high	0.8000	0.9	low	0.7700	0.9	low	0.2400	0.7	low	0.9600	1.0	low			
UCS	Overall survival	56	TCGA	0.0360	0.5	low	0.3700	1.4	high	0.5100	0.8	low	0.3700	1.4	high	0.6400	0.9	low			
UVM	Overall survival	78	TCGA	0.5300	1.3	high	0.1300	2.0	high	0.0210	2.9	high	0.4300	0.7	low	0.0410	2.5	high			

Abbreviations of cancer type as: Adrenocortical Cancer (ACC), Bladder Cancer (BLCA), Breast Cancer (BRCA), Cervical Cancer (CESC), Bile Duct Cancer (CHOL), Colon Cancer (COAD), Colon and Rectal Cancer (COADREAD), Large B Cell Lymphoma (DLBC), Mesothelioma (MESO), Esophageal Cancer (ESCA), Glioblastoma (GBM), Kidney Chromophobe (KICH), Kidney Clear Cell Carcinoma (KIRC), Kidney Papillary Cell Carcinoma (KIRP), Acute Myeloid Leukemia (LAML), Lower Grade Glioma (LGG), Liver Cancer (LIHC), Lung Adenocarcinoma (LUAD), Lung Squamous Cell Carcinoma (LUSC), Head and Neck Cancer (HNSC), Ovarian Cancer (OV), Pancreatic Cancer (PAAD), Pheochromocytoma and Paraganglioma (PCPG), Prostate Cancer (PRAD), Rectal Cancer (READ), Melanoma (SKCM), Stomach Cancer (STAD), Testicular Cancer (TGCT), Thyroid Cancer (THCA), Thymoma (THYM), Endometrioid Cancer (UCEC), Uterine Carcinosarcoma (UCS), Ocular Melanoma (UVM). All data collected and manipulated from UCSC Xena website (https://xenabrowser.net/, accessed on 15 September 2020).

**Table 3 biomedicines-09-01159-t003:** Functions of *LGALS* in cancer.

Cancer Type	*LGALS* Expression	Biological Relevance	Year	Author	Reference
Bladder	Gal-1	Regulation of proliferation and invasion	2018	Li, C.F.	[66]
	Gal-3	Tumor Growth	2008	Fang, T.	[67]
	Gal-9	Contribution to tumor invasion and immune surveillance	2019	Qi, Y.	[68]
Blood	Gal-12	Regulation of lipid raft formation	2016	Xue, H.	[69]
Bone marrow	Gal-1	Regulation of M2 macrophage activation	2017	Andersen, M.N.	[70]
	Gal-1	Required for tumor development	2019	Muller, J.	[71]
Breast	Gal-1	Contributes to tumor progression and drug resistant	2017	Nam, K.	[72]
	Gal-1	Tumor metastasis and immune evasion	2019	Patrick, M.E.	[73]
	Gal-1	Associated with chemoresistance	2016	Upreti, M.	[74]
	Gal-3	Associated with metastasis	2019	Pereira, J.X.	[75]
	Gal-3	Involved in osteoclastogenesis	2016	Nakajima, K.	[76]
	Gal-3	Regulates tumor growth and metastasis	2003	John, C.M.	[77]
Cervical	Gal-1	promotes the invasive	2020	Wu, H.	[63]
	Gal-7	Negative regulation of tumor progression	2016	Higareda-Almaraz, J. C.	[78]
Colon	Gal-1	Promotes invasion	2017	Park, G.	[79]
	Gal-3	Promotes cancer metastasis	2013	Dovizio, M.	[80]
	Gal-12	Inhibits glutamine uptake	2019	Katzenmaier, E.M.	[81]
Colorectal	Gal-1	Associated to immunosuppressive	2021	Cagnoni, A.J.	[43]
	Gal-1	Associated to tumor progression	2019	Sandberg, T.P.	[82]
	Gal-3	Promotes metastasis	2017	Wu, K.L.	[56]
	Gal-4	causing apoptosis	2017	Rao, U.	[42]
	Gal-4	Growth inhibition	2019	Michalak, M.	[83]
	Gal-9	Increases immune surveillance	2019	Sakhnevych, S.S.	[84]
Esophageal	Gal-9	Apoptosis inductor	2019	Chiyo, T.	[85]
Gastric	Gal-1	Involved in metastasis	2016	Chong, Y.	[86]
Head and neck	Gal-1	Regulates vessel normalization	2017	Koonce, N.A.	[87]
Liver	Gal-3	Regulates cell proliferation	2021	Liang, Z.	[44]
	Gal-9	Promotes tumor apoptosis	2017	Tadokoro, T.	[88]
Lung	Gal-1	Correlated to metabolism and poor prognosis	2019	Zheng, H.	[89]
	Gal-1	Immune suppression	2016	Hsu, Y. L.	[90]
	Gal-3	Associated with drug-resistance	2019	He, F.	[91]
	Gal-3	Immune surveillance escape	2019	Vuong, L.	[92]
	Gal-9	Associated with chemoresistance	2019	Limagne, E.	[51]
Ovarian	Gal-3	Chemotherapy sensitivity	2019	Wang, D.	[93]
	Gal-3	Cell apoptosis	2016	El-Kott, A.F.	[94]
	Gal-3	Cell motility and sphere formation	2019	Hossein, G.	[95]
Pancreatic	Gal-1	Promotes cancer progression	2018	Tang, D.	[96]
	Gal-1	Crosstalk with stromal cells	2018	Orozco, C. A.	[97]
Prostate	Gal-1	Associated to invasion ability	2018	Shih, T.C.	[48]
	Gal-1	Regulates cells proliferation and apoptosis	2018	Corapi, E.	[98]
	Gal-3 (cytoplasmic)	Promotes tumor progression	2004	Califice, S.	[14]
	Gal-3 (nucleus)	Inhibits tumor progression	2004	Califice, S.	[14]
	Gal-3	Immunosuppressive and Metastasis	2020	Caputo, S.	[50]
	Gal-3	Regulates osteoclastogenesis	2016	Nakajima, K.	[76]
Skin	Gal-1	Involved in immune surveillance escape and cause drug resistance	2020	Gorniak, P.	[99]
	Gal-3	Lung metastasis	2005	Krishnan, V.	[100]

All the biological relevancies of Gals in cancer are collected and listed based on current research and referred to in this article.

**Table 4 biomedicines-09-01159-t004:** Interaction partners of galectin.

***LGALS* Type**	**Protein Partner**	**Dataset**
*LGALS1*	AAR2, ACO, AGR2, ALCAM, ALDOA, ANXA2, ANXA22, APEX, APOA1, ARF4, AFP, ATP6AP2, C110RF87, CD2, CD4, CD7, CD28, CD44, CD68, CDC42, CDHR2, CDHR5, CHL1, CHORDC1, CLNS1A, CRIP1, CYLD, DBN1, DCPS, DDX19B, DYRK1A, EFNB3, EFTUD2, EGFR, EREG, ESR2, F5, FGA, FGG, FAM24B, FLNA, FN1, FUBP1, FZD10, GEMIN4, GOLT1B, GTF2I, HEPACAM2, HIST1H2BO, HNRNPL, HRAS, HSPA5, HSPB2, ICAM2, ICOSLG, IGBP1, IL6, ITGA4, LAMC1, LAMP1, LAMTOR3, LASP1, LGALS3BP, LGALS3, LIMA1, LINGO2, LMAN1, LRFN4, MB21D1, MCM2, MCM5, MUC16, MYC, MYH9, NLGN3, NPM1, NTRK3, PCBD2, PCBP1, PCBP2, PECAM1, PHB2, PIGR, PIH1D1, PLIN3, POLE4, PRKCZ, PSG1, PSMG1, PTEN, PTGER3, PTPRA, PTPRC, PTPRZ1, RAB5C, RAB10, RAC1, RAE1, RNF4, SIPR2, SEMA4C, SERPINH1, SIGLEC7, SLAMF1, SLAMF7, SMN1, SNRPB, SOD1, SOX2, SPANXC, SPN, STUB1, SUSD2, TALDO1, TIMP1, TNFRSF10C, TNF, TYW3, U2AF2, UBE2N, UQCRFS1, USP4, VASN, VCAM1, VIPR2, WWP2, ZNF131	Abbott KL(2008), Agrawal P(2010), Amith SR (2017), Byron A(2012), Caron P(2019), Chen R(2013), Cho Y(2018), Chung LY (2012), Drissi R (2015), Elliott PR (2016), Ewing RM (2007), Fang X (2011), Foerster S (2013), Giurato G (2018), Grose JH (2015), Guo HB (2009), Havugimana PC (2012), Heidelberger JB (2018), Hein MY (2015), Hou C (2018), Humphries JD (2009), Hutchins JR (2010), Huttlin EL (2014/pre-pub), Huttlin EL (2015), Huttlin EL (2017), Kristensen AR (2012), Kumar R (2017), Kupka S (2016), Lin TW (2015), Lum KK (2018), Malinova A (2017), Pace KE (1999), Park JW (2001), Paz A (2001), Roewenstrunk J (2019), Seelenmeyer C (2003), Shen C (2015), Tiemann K (2018), Tinari N (2001), Varier RA (2016), Verrastro I (2015), Voss PG (2008), Walzel H (2000), Wan C (2015), Wang J (2008), Whisenant TC (2015), Yamauchi T (2018), Zhao B (2012)
*LGALS2*	ALOX5AP, APP, GCSAM, IGSF23, IKBKG, LTA, NAT8, NR1H4, NXPE1, PAICS, PSMA6, SDCBP2, SDCBP, SDPR, TRIM16, TUBA1B, TUBB, WDYHV1	Chauhan S (2016), Fenner BJ (2010), Luck K (2020), Ozaki K (2004), Rolland T (2014)
*LGALS3*	ORF3A, ORF7B, ABCB1, ABCB1, ABCC4, ABCC4, ACAA2, ACOT1, ACP2, ACTA1, ACTA2, ADCY3, ADCY3, ADCY6, ADCY6, ADCY9, ADCY9, AGPS, AGR2, AHCY, AK3, ALCAM, ALCAM, APLNR, APP, ATG9A, ATG9A, ATP1A1, ATP2B4, ATP2B4, ATP5C1, ATP13A3, ATP13A3, BARD1, BARD1, BARD1, BARD1, BCL2L14, BRCA1, BSG, C1GALT1C1, C1ORF85, C11ORF87, CACNG1, CACNG5, CADM1, CAPN1, CAPN1, CBFB, CCT3, CD58, CD58, CD63, CD63, CD68, CD109, CD109, CDC5L, CDK15, CFTR, CKAP4, CLCN3, CLCN3, CLCN5, CLEC7A, CLNS1A, COASY, COASY, COLEC12, COLEC12, COX17, CRIP1, CRX, CRYZ, CSPG4, CSPG4, CTNNB1, CYLD, DBI, DDOST, DSTN, DUT, ECE1, ECE1, EGFR, EGFR, EGFR, ELMOD1, ELN, ELN, EMB, EMB, EMP3, ENO1, ENPP4, ENPP4, ESR1, ESR2, FAHD2A, FBXL4, FBXO6, FCF1, FCGR2A, FKBP2, FLT4, FLT4, FN1, FN1, FOXA1, GEMIN4, GLB1, GOLGA2, GPR35, GPR35, GPR52, GPR52, GPR55, GPR55, GPR84, GRPR, GSTP1, GTF2I, GTF2I, GTF2I, HEBP1, HEG1, HEG1, HRAS, HRNR, HSP90AB1, HTR2C, ICOSLG, IDH2, IGBP1, IL31RA, IMPA2, INCA1, IPO5, IPPK, ISOC2, ITGA1, ITGA1, ITGA2, ITGA2, ITGB1, KCTD12, KIAA0319L, KIAA0319L, KIAA1549, KIAA1549, KIAA1549, KIF16B, KPNB1, KRAS, LAMA1, LAMA1, LAMA4, LAMA4, LAMB1, LAMB1, LAMC1, LAMP1, LAMP1, LAMP2, LGALS1, LGALS3BP, LGALS3BP, LGALS3BP, LGALS3BP, LGALS3BP, LGALS3BP, LGALS3, LGALS9C, LGALS9C, LGALS9, LGALS9, LNPEP, LNPEP, LPAR1, LPAR1, LPAR2, LRP1, MAP1LC3A, MAP1LC3A, MAPK3, MCCC1, MCCC2, MCPH1, MEFV, MFAP3, MFAP3, MFSD4, MPZL1, MPZL1, MRC2, MRC2, MSRB2, MUC, MYH10, MYL6, MYL12B, MYO1E, MYOC, NAP1L1, NCSTN, NHLRC2, NID2, NID2, NPTN, NRAS, OPALIN, OPTN, OSTM1, OSTM1, P2RY6, P2RY12, PARK7, PARP1, PAXIP1, PAXIP1, PCBP2, PCCA, PDCD1LG2, PDHX, PEBP1, PKM, PLIN3, PODXL, PODXL, PPARG, PPIA, PPIG, PPP2R1A, PRDX5, PRPF8, PRPF19, PRR13, PTGFRN, PTGFRN, PTPN11, PTPN11, PTPRH, PTPRJ, PTPRJ, PTPRK, PTPRK, PTPRO, PTPRZ1, PTPRZ1, PYHIN1, PYHIN1, RAB7A, RAB11B, RNF167, RPL7A, RPL12, RPL35, RPS20, RRAGB, RRAGB, RRAGC, RRAGC, RTN4RL2, RTN4RL2, RUNX1, SCARA3, SCARA3, SDK1, SDK1, SDK2, SDK2, SERINC1, SERINC2, SERPINH1, SGOL2, SH3BGRL, SLC1A3, SLC1A5, SLC1A5, SLC2A14, SLC3A2, SLC4A2, SLC4A2, SLC4A7, SLC4A7, SLC5A5, SLC5A8, SLC6A1, SLC7A2, SLC7A2, SLC9A6, SLC9A6, SLC12A2, SLC12A2, SLC12A2, SLC12A4, SLC12A4, SLC12A6, SLC12A6, SLC12A7, SLC12A7, SLC14A1, SLC15A4, SLC15A4, SLC17A5, SLC25A5, SLC26A2, SLC26A2, SLC30A1, SLC30A1, SLC38A9, SLC38A9, SLC46A3, SNCA, SOD1, SOD2, SPR, SQSTM1, SS18L1, SUFU, SYPL1, TACR1, TBK1, TEX35, TEX35, TKT, TMEM9, TMEM63A, TMEM63A, TMEM63C, TMEM179B, TMEM182, TMPO, TMSB10, TPCN2, TRIM5, TRIM5, TRIM6, TRIM16, TRIM16, TRIM17, TRIM22, TRIM23, TRIM49, TRIM49, TSPAN2, TSPAN31, TXN, VIM, VSTM1, YWHAZ	Bussi C (2018), Carvalho RS (2014), Chauhan S (2016), Chen X (2014), Cortegano I (2000), Elliott PR (2016), Fautsch MP (2006), Foerster S (2013), Giurato G (2018), Guo HB (2009), Havugimana PC (2012), Hein MY (2015), Hubel P (2019), Huttlin EL (2014/pre-pub), Huttlin EL (2015), Huttlin EL (2017), Jozwik KM (2016), Ju T (2008), Koths K (1993), Kumar P (2017), Li CG (2019), Li X (2016), Lin TW (2015), Liu X (2017), Luck K (2020), Malinova A (2017), Merlin J (2011), Mohammed H (2013), Moutaoufik MT (2019), Ochieng J (1999), Olah J (2011), Park JW (2001), Rolland T (2014), Rosenberg I (1991), Rosenbluh J (2016), Stukalov A (2020), Tiemann K (2018), Tinari N (2001), Ulmer TA (2006), Voss PG (2008), Wan C (2015), Wang X (2006), Woods NT (2012), Yeung ATY (2019), Zhong Y (2017)
*LGALS4*	ARPC3, BANP, CEP55, CHD7, CFTR, ELAVL1, ELAVL2, ELAVL3, ELAVL4, EYA2, GOLGA6L9, HOMEZ, HOXA1, HSF2BP, KRTAP11-1, MLH1, NOTUM, NCKIPSD, PAICS, RFX6, PHGDH, RRAS2, SHKBP1, SPANXA1, TENM4, TOX2	Wang X (2006), Luck K (2020), Brieger A (2010), Stelzl U (2005)
*LGALS7*	ADSS, AEBP2, AHSG, APAF1, CBWD1, CDK2, CHD3, CHD4, CHST6, COPS5, CYLD, DDX19B, DOCK7, DPYSL2, DPYSL4, DPYSL5, E2F6, ESR1, EZH2, FGFR1, GAB1, HECW2, HIV2GP4, HNRNPA1, HRAS, HSPA6, IFI16, INSIG2, KLHL8, LGALS7B, LSM2, LUCAT1, LZTR1, MCM2, METTL3, MKS1, MTPAP, MYC, NAMPT, PALD1, PCGF5, PIK3C3, PKN2, POLA2, POLE2, PPARG, PRPF8, RHEB, ST3GAL1, STG3GAL2, ST3GAL4, ST6GALNAC2, ST6GALNAC3, ST6GALNAC4, ST8SIA6, SSBP1, SUZ12, TAB1, TCF3, TGM5, TUBA1B, TUBA3C, TUBA4A, TUBB3, TUBBB, TUBG1, USP1, USP4, USP15, USP38, VPS13B, VWA9, WIPI1, XPO1, YAF2	Hauri S (2016), Adhikari H (2018), Behrends C (2010), Bennett EJ (2010), Cao Q (2014), Diner BA (2015), Drissi R (2015), Elliott PR (2016), Fogeron ML (2013), Guardia-Laguarta C (2019), Hauri S (2016), Heidelberger JB (2018), Hoffmeister H (2017), Huttlin EL (2014/pre-pub), Huttlin EL (2017), Ikeda Y (2009), Kirli K (2015), Landsberg CD (2019), Li CG (2019), Lu L (2013), Luck K (2020), Malinova A (2017), Neganova I (2011), Pladevall-Morera D (2019), Roy R (2014), Sowa ME (2009), Tarallo R (2011), Teachenor R (2012), Zhou Q (2019)
*LGALS8*	ABCC1, ABCC4, ACP2, ADCY6, ADCY9, ALCAM, ANO6, APEH, ATG9A, ATP2B2, ATP2B3, ATP2B4, ATP6VOA1, ATP6VOA2, ATP13A3, BARD1, C10RF85, C10RF159, C4A, CALCOCO2, CD47, CD58, CD63, CD276, CLCN3, CLCN5, CLCN7, COLEC12, CSAD, CSPG4, CUL1, DAG1, ECE1, EGFR, ELTD1, EMB, ENPP4, ESR2, FGFR1, FLE4, FN1, HEG1, ITGA1, ITGA2, ITGA3, ITGA5, ITGA6, ITGA7, ITGB1, KIAA0319L, KIAA1549, LAMA4, LNPEP, LRRC4B, LRRC8A, LRRC8C, LRRC8E, LRRK2, LRSAM1, MCAM, MEFV, MFAP3, MID2, MRC2, NACC1, NCR3LG1, NDP, NPC1, NPTN, NR1D2, OPTN, OSTM1, PAN2, PDPN, PHACTR1, PODXL, PROCR, PTGFRN, PTPRA, PTPRG, PTPRJ, PTPRK, PVRL1, PVRL3, RNF13, RRAGB, RRAGC, SCARB2, SDCBP, SDK1, SEZ6L2, SLC1A1, SLC4A2, SLC4A7, SLC9A6, SLC12A2, SLC12A4, SLC12A6, SLC12A7, SLC12A9, SLC17A5, SLC22A23, SLC26A2, SLC30A1, SLC31A1, SLC35A5, SLC36A1, SLC38A9, SORL1, SPPL2A, SUSD5, SV2A, TAX1BP1, TMEM63A, TMEM63B, TMEM237, TMEM242, TRIM5, TRIM6, TRIM17, TRIM22, TRIM23, TRIM49, TRPM4, WBP2	Beilina A (2014), Bennett EJ (2010), Bett JS (2013), Chauhan S (2016), Chen S (2018), Foerster S (2013), Giurato G (2018), Hadari YR (2000), Huttlin EL (2014/pre-pub), Huttlin EL (2015), Kim BW (2013), Li S (2013), Luck K (2020), Rolland T (2014), Rual JF (2005), Thurston TL (2012), Tomkins JE (2018), Verlhac P (2015), Woods NT (2012)
*LGALS9*	ACP2, ALCAM, ATG9A, ATG16L1, ATP2B4, ATP7A, ATP11C, ATP13A3, C2CD5, CD44, CD47, CD58, CD63, CD86, CD109, CD274, CD276, CLCN3, COLEC12, CSPG4, CTLA4, CTNNB1, DAG1, DAZAP2, ECE1, ENPP4, ENPP4, FBLN1, FN1, FOXP3, HAVCR2, IGF2R, IGSF3, ITGA4, ITGB1, JUP, KIAA0319L, KIAA15459, KRTAP6-3, LAG3, LAGALS3, LGALS9B, LGALS9C, LAMA1, LAMA4, LAMB1, LGALS3, LGALS9B, LGALS9C, LNPEP, LRP1, LUM, MAN2B1, MB21D1, MET, MFAP3, MPZL1, MRC2, NAGLU, NCR3LG1, NICD1, NICD2, NR2C2, OSTM1, P4HB, PBK, PCDH9, PDIA3, PDIA6, PLXNA1, PODXL, PTGFRN, PTPN11, PTPRJ, PTPRK, RNF13, RRAGB, RRAGC, SLC1A5, SLC4A7, SLCSLC6A6, SLC9A3R2, SLC9A6, SLC12A2, SLC12A4, SLC12A6, SLC12A7, SLC12A9, SLC29A1, SLC38A9, SORL1, SUSD5, TSPAN3, USP39, ZER1	Arbogast F (2019), Bi S (2011), Ewing RM (2007), Huttlin EL (2014/pre-pub), Huttlin EL (2015), Luck K (2020), Lum KK (2018), Wan C (2015), Yu L (2018)
*LGALS10*	ADH1A, CHD7, CLC, EPX, GALE, ISYNA1, LGALS3, LGALS12, PAICS, RNASE2, RNASE3	Rolland T (2014), Luck K (2020)
*LGALS12*	CLC, EMB, HAPLN1, LGALS13, LIMD1, RYR3, SLC7A2, VPS12C, VPS13C	Huttlin EL (2015), Huttlin EL (2017)
*LGALS13*	ADAM12, ANXA2, BTBD1, CENPV, CREB5, DNPEP, ENG, ENDOU, HOXA1, LGALS3, LGALS12, NUTF2, NUFIP2, OTX1, PACSIN3, PAPPA, PGF, PHLDA1, POLR1A, POU4F2, PPEF1, PWP1, UBB	Huttlin EL (2017), Lambert B (2012), Luck K (2020), Rolland T (2014), Yu H (2011)
*LGALS14*	ADAMTSL4, AJUBA, ALS2CR12, BANP, BLZF1, C1QTNF2, COG6, CRLF3, DDIT4L, DOK6, ENTHD2, FCHO1, FLNA, GFNA, GFAP, IKZF3, IL16, ITLN2, JRK, KRT19, KRT35, LNX1, LNX2, MEI4, MID2, NFKBID, PCBID, PCBD2, PICK1, POF1B, PPM1J, REL, RIMBP3C, SDCBP2, SH3GLB2, SPAG5, STAT5B, TARSL2, TBC1D25, TCF4, TEKT1, TNR, TRIM23, TRIM27, VIM, ZBTB8A, ZNF71, ZNF248, ZNF438, ZNF558, ZNF655	Luck K (2020), Rolland T (2014), Yachie N (2016)

The relative interaction proteins of Gals were collected based on the BioGRID website (https://thebiogrid.org/, accessed on 15 September 2020).

**Table 5 biomedicines-09-01159-t005:** Subcellular distribution of galectin.

*LGALS* Type	Nucleus	Endoplasmic Reticulum	Cytosol	Plasma Membrane	Extracellular	Cytoskeleton	Mitochondrion	Peroxisome	Endosome	Lysosome	Golgi Apparatus
*LGALS1*	+	+	+	+	+	+	+	+	+	+	+
*LGALS2*	+	+	+	+	+	+	+				
*LGALS3*	+	+	+	+	+	+	+	+	+	+	+
*LGALS4*	+	+	+	+	+	+	+	+	+	+	+
*LGALS7*	+	+	+	+	+	+	+	+	+	+	+
*LGALS8*	+	+	+	+	+	+	+	+	+	+	+
*LGALS9*	+	+	+	+	+	+	+		+	+	+
*LGALS10*	+	+	+	+	+	+	+	+	+	+	+
*LGALS12*	+	+	+	+	+	+	+			+	+
*LGALS13*	+	+	+	+	+	+	+			+	+
*LGALS14*	+	+	+	+	+		+				+

+ Means positive expressed. Listed are the distributions of Gals in cell subcellular components, which were collected from the GeneCards website (https://www.genecards.org/, accessed on 15 September 2020).

**Table 6 biomedicines-09-01159-t006:** Galectin-related ongoing clinical trials.

Drug	Target	Phase	Status	Disease	NCT Number
GCS-100	Gal-3	I	Completed	Chronic Kidney Disease	NCT01717248
		II	Completed	Chronic Kidney Disease	NCT01843790
		II	Withdrawn	Diffuse Large B-cell Lymphoma	NCT00776802
GM-CT-01	Gal-3	II	Terminated	Metastatic Melanoma	NCT01723813
		II	Withdrawn	Colorectal Cancer	NCT00388700
		II	Withdrawn	Cancer of the Bile Duct, Gallbladder Cancer	NCT00386516
		II	Terminated	Colorectal Cancer	NCT00110721
GR-MD-02	Gal-3	I	Completed	Metastatic Melanoma	NCT02117362
		I	Recruiting	Melanoma, Non-Small Cell Lung Cancer, Squamous Cell Carcinoma of the Head and Neck	NCT02575404
		I	Completed	Non-Alcoholic Steatohepatitis (NASH)	NCT01899859
		II	Completed	Hypertension, Portal	NCT02462967
		II	Recruiting	Prevention of Esophageal Varices, NASH - Nonalcoholic Steatohepatitis, Cirrhosis	NCT04365868
		II	Completed	Nonalcoholic Steatohepatitis	NCT02421094
		II	Completed	Psoriasis	NCT02407041
OTX008	Gal-3	I	Unknown	Solid Tumors	NCT01724320
PectaSol-C Modified Citrus Pectin (MCP)	Gal-3	II	Completed	Prostatic Neoplasms	NCT01681823
TD139	Gal-3	II	Recruiting	Idiopathic Pulmonary Fibrosis (IPF)	NCT03832946
		II	Completed	Idiopathic Pulmonary Fibrosis	NCT02257177
		II	Recruiting	COVID-19	NCT04473053

All data were collected and manipulated from the NIH ClinicalTrials website (https://clinicaltrials.gov/, accessed on 15 September 2020).

## Data Availability

Related available websites for datasets generated in this article were noted.

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
