# Peer review of "Galectins in Cancer and the Microenvironment: Functional Roles, Therapeutic Developments, and Perspectives"

_biomedicines, 2021, doi:10.3390/biomedicines9091159_

Round 1

Reviewer 1 Report

The review article is comprehensive, but some details must be improved:

Legend to Fig 1. The web of protein atlas must be included to the text,, the same abou other web sources

Table 1. The source?

Table 2+3.Characters are to small

Table 4. Characters are to small

Table 4. Characters are to small, Golgi not golgu - referencs are necessary

Author Response

Reviewer: 1

The review article is comprehensive, but some details must be improved:

  • Ans: We thank the Referee for the time taken reviewing our work and for the constructive comments.

Legend to Fig 1. The web of protein atlas must be included to the text, the same about other web sources

  • Ans: We thank the Reviewer for bringing up this important point. We agree with the reviewers' comments. As requested by the Referee, we have included the resource in each Figures and Tables.

Please refer to the Figure.1 lines 70 to 71;

Please refer to the Table.1 lines 74;

Please refer to the Table.2 lines 171;

Please refer to the Table.4 lines 218.

Please refer to the Table.5 lines 251.

Please refer to the Table.6 lines 422.

Table 1. The source?

  • Ans: We thank the Reviewer for pointing out the importance. We agree with the reviewers' comments. As requested by the Referee, we have included the resource in each Figures and Tables.

Please refer to the Table.1 lines 73 to 74.

Table 2+3.Characters are to small

  • Ans: We apologize for our small characters. We agree with the reviewers' comments. As requested by the Referee, we have now replaced Tables in as enlarged characters as possible.

Please refer to the Table.2 lines 159 to 171;

Please refer to the Table.3 lines 172 to 179.

Table 4. Characters are to small

  • Ans: We apologize for our small characters. We agree with the reviewers' comments. As requested by the Referee, we have now replaced Tables in as enlarged characters as possible.

Please refer to the Table.4 lines 215 to 218.

Table 4. Characters are to small, Golgi not golgu - referencs are necessary

  • Ans: We apologize for our small characters. We agree with the reviewers' comments. As requested by the Referee, we have now corrected words and replaced Tables in as enlarged characters as possible.

Please refer to the Table.5 lines 248 to 251.

We thank the Reviewer for this important observation and related comments as well as professional review work on our manuscript. We hope that our revised manuscript is acceptable for publication in Biomedicines.

Reviewer 2 Report

This work by Chien-Hsiu Li on the contribution of galectins to cancer development is challenging, because there is a wealth of data to cover. The authors finally arrive to summarize well in a nicely written and structured review.

Minor remarks:

A figure on the structure of the different galectins could help.

There is an important review article on the use of galectins as biomarkers in different cancers (Thyssen et al, Biochim Biophys Acta. 2015) that merits a citation.

Author Response

Reviewer: 2

This work by Chien-Hsiu Li on the contribution of galectins to cancer development is challenging, because there is a wealth of data to cover. The authors finally arrive to summarize well in a nicely written and structured review.

  • Ans: We deeply appreciate this Reviewer for their positive and insightful suggestion.

Minor remarks:

A figure on the structure of the different galectins could help.

  • Ans: We thank this Reviewer for their positive and constructive suggestion. We agree with the reviewers' comments. As requested by the Referee, we have included a new Figure on the structure of the different galectins.

Please refer to the Figure.1 lines 67 to 71.

There is an important review article on the use of galectins as biomarkers in different cancers (Thyssen et al, Biochim Biophys Acta. 2015) that merits a citation.

  • Ans: We thank the Reviewer for pointing this out to us. To address the Reviewer's concern, we have now included references to highlight the related works of the senior author's lab.

Please refer to the lines 31 to 33.

We thank the Reviewer for this important observation and related comments as well as professional review work on our manuscript. We hope that our revised manuscript is acceptable for publication in Biomedicines.

Round 2

Reviewer 1 Report

I think that revised manuscript can be published